# Reference values of normal fetal ductus venosus Doppler flow measurements at 11–14 weeks of gestation

**Shizhen Li** [ID]**, Haifang Wu, Linlin Zhu, Qi Li, Xiangyi Dong** [ID]*

Department of Ultrasound, Qilu Hospital of Shandong University, Jinan, China

* dongxy907@163.com

## Abstract

### Objectives

To establish the reference range of normal fetal ductus venosus pulsatility index (DV PI) and ductus venosus (DV) blood flow velocity at 11–14 weeks of gestation.

### Material and methods

Fetal ductus venosus Doppler flow was measured in singleton pregnancies attending our hospital for early pregnancy nuchal translucency (NT) screening between June 2021 and May 2022. All fetuses were followed up for pregnancy outcome using the following inclusion criteria: Singleton pregnancy; no maternal underlying diseases such as diabetes, hypertension, rheumatism, or other pregnancy complications; fetal crown-rump length (CRL) of 45 to 84 mm; normal NT screening ultrasound; no absent or reversed ductus venosus a-wave; no fetal structural abnormalities; no chromosomal abnormalities during follow-up; and good pregnancy outcome. DV PI, peak ventricular systolic velocity (S-wave), atrial systolic flow velocity (a-wave) and time-averaged maximum velocity (TAMXV) were recorded.

### Results

The ductus venosus Doppler parameters of 224 fetuses which met the inclusion criteria were analysed. DV PI P5 and P95 ranged from 1.0007 and 1.3415 for a CRL of 45 mm to 0.9734 and 1.2115 for a CRL of 84 mm, indicating a statistically significant correlation with CRL. DV S-wave, a-wave, and TAMXV all increased as CRL increased, demonstrating a statistically significant correlation with CRL values.

### Conclusions

A reference range of normal fetal ductus venosus Doppler spectral parameters at 11–14 weeks was established to provide a basis for further research into the clinical value of normal and abnormal DV PI values in relation to adverse pregnancy outcomes.

**Data Availability Statement:** All relevant data are within the manuscript and its Supporting information files.

**Funding:** The author(s) received no specific funding for this work.

**Competing interests:** The authors have declared that no competing interests exist.

## Introduction

The ductus venosus (DV) is an important, funnel-like blood circulation conduit located between the portal sinus (PS) and the inferior vena cava (IVC) subdiaphragmatic vestibulum, which is unique to fetal circulation [1]. Highly oxygenated blood from the placenta is preferentially supplied to vital organs such as the heart and brain of the fetus via the DV [2]. The anatomy of the DV reflects the pressure gradient between the umbilical vein and the right atrium, and the measurement of its blood waveform indirectly reflects the heart function and health condition of the fetus. The DV blood flow spectrum includes the peak ventricular systolic velocity (S-wave), ventricular diastolic peak (D-wave), and atrial systolic flow velocity (a-wave). This is typically characterized by a consistent antegrade flow throughout the cardiac cycle with no inversion during atrial systole. The evaluation of the DV blood flow spectrum mainly includes qualitative evaluation (lack of the a-wave or reverse a-wave) and quantitative evaluation (DV pulsatility index, PI). Several studies have shown that abnormalities in the PI at 11–14 weeks are strongly associated with chromosomal abnormalities, congenital heart malformations and adverse pregnancy outcomes [3–6]; therefore, it is a useful tool for assessing fetal abnormalities during early pregnancy. At present, there are only a few clinical applications of DV PI values and a lack of studies determining normal reference values. The aim of this study was to establish the reference range of DV PI and DV blood flow velocity for normal fetuses in transabdominal ultrasound examination at 11–14 weeks of gestation. This will facilitate guidance on antenatal consultation and management, and provide a scientific basis for future studies on DV waveforms and DV PI values.

## Methods

### Study participants

Fetal ductus venous Doppler flow was measured in singleton pregnancies presenting to our hospital (A tertiary teaching hospital with prenatal diagnosis qualification located in Jinan, China) for NT screening in early pregnancy from June 1, 2021 to May 31, 2022, and all fetuses were followed up for pregnancy outcome. This study was approved by the ethics committee of Qilu Hospital of Shandong University (No.2019209) and all patients gave their written informed consent; their anonymous information will be published in this article.

Inclusion criteria were: Singleton pregnancy; crown-rump length (CRL) of 45–84 mm; normal nuchal translucency (NT); no maternal disease such as diabetes mellitus, hypertension, rheumatism, or other pregnancy complications; no missing or reversed ductus venosus a-wave; no fetal structural malformations or chromosomal abnormalities during follow-up; and good pregnancy outcome.

### Doppler ultrasonography

All examinations are performed by a British Fetal Medicine Foundation (FMF) certified ultrasonographer with over 10 years of experience in fetal ultrasonography. The ultrasound instrument was a PHILIPS EPIQ7 color Doppler ultrasonography with a C5-1 transabdominal convex array probe.

All examinations were performed transabdominally with the fetus in a quiet state. Firstly, a median sagittal view of the fetal trunk was obtained and the CRL and NT were measured. Then a parasternal median sagittal view of the right side of the fetal thorax and abdomen was obtained, and the image was enlarged until the fetal thorax and abdomen occupied the entire image. The umbilical vein, portal sinus, DV, and fetal heart were displayed by color Doppler [7]. The brightest part of the blood flow distal to the PS was the DV, where the pulsed Doppler

sampling frame was placed. the sampling gate was adjusted by 0.5–1 mm to avoid contamination from the PS, left hepatic vein, and inferior vena cava [8]. The angle between the sampling line and the blood flow beam was <30˚ and at least four consistent regular Doppler waveforms were obtained. DV PI, S-wave, a-wave, and time-averaged maximum velocity (TAMXV) were measured automatically, repeated three times, and the mean value was taken and recorded.

### Statistical analysis

Statistical analysis was performed using SPSS software (IBM SPSS Statistics for Windows, version 22.0. Armonk, NY: IBM Corp). The Kolmogorov-Smirnov normality test was first applied to the CRL, PI, S-wave, a-wave, and TAMXV data. If the variables were not normally distributed, a Box-Cox transformation was applied to make them normally distributed. Pearson's correlation was used to analyse the significance of the four variables PI, S-wave, a-wave, and TAMXV in relation to CRL. The mean predictive values (MEAN) of the four variables were calculated for different CRLs. A polynomial regression model was established with the absolute values of the residuals of the regression model for each of the four variables as the dependent variable and CRL as the independent variable. The standard deviations (SD) of the four variables were estimated by multiplying the predicted value of the residual model by $\sqrt{\pi/2} = 1.2534$. The lower limit of the 90% reference range was $P_5 = \text{MEAN} - 1.6449 \times \text{SD}$, and the upper limit was $P_{95} = \text{MEAN} + 1.6449 \times \text{SD}$.

## Results

We retrospectively analysed 261 fetuses, of which 10 were excluded for maternal reasons, 19 for NT thickening, chromosomal abnormalities or fetal malformations detected by ultrasound in mid to late pregnancy, 1 for intrauterine death, 1 for induction of labor for unknown reasons, 6 for missing or reversed DV a-waves which displaying distinct differences in distribution in comparison to normal fetuses. Finally, we included 224 fetuses, all of whom met the inclusion criteria. The average maternal age was 30.2 years, the average body mass index was 22.5, the median number of pregnancies was 2, and the median number of deliveries was 0.

For the 224 normal fetuses, the DV PI P5 and P95 ranged from 1.0007 and 1.3415 for a CRL of 45 mm to 0.9734 and 1.2115 for a CRL of 84 mm. This correlation with CRL values was statistically significant (S1 Fig; Pearson's r = -0.163; P = 0.015). The S-wave P5 and P95 ranged from 39.6789 and 65.5636 cm/s for a CRL of 45 mm to 52.0146 and 66.9851 cm/s for a CRL of 84 mm (S2 Fig; Pearson's r = 0.167; P = 0.013). The a-wave P5 and P95 ranged from 6.2287 and 16.6637 cm/s for a CRL of 45 mm to 9.8525 and 14.9860 cm/s for a CRL of 84 mm (S3 Fig; Pearson's r = 0.162; P = 0.015). The TAMXV P5 and P95 ranged from 25.5151 and 46.1257 cm/s for a CRL of 45 mm to 35.5750 and 49.8634 cm/s for a CRL of 84 mm (S4 Fig; Pearson's r = 0.204; P < 0.002). All three blood flow velocities increased with CRL, with statistically significant correlations (Table 1).

## Discussion

Qualitative DV assessment has now become an important part of NT screening in early pregnancy, and the assessment of whether a-waves are missing or reversed has improved the identification rate of trisomy 21 [9]. Clinical use of quantitative assessments of the DV (e.g., DV PI) are relatively infrequent, while studies have shown that using the 95th percentile of DV PI as a threshold outperforms qualitative assessment of the DV in detecting chromosomal abnormalities [3, 4, 10, 11]. Abnormal DV blood flow may be an independent marker of fetal congenital heart disease (CHD) [12–15]. Timmerman et al. [5] found a higher detection rate of

**Table 1. DV PI, S-wave, a-wave, TAMXV 5th and 95th percentile measurements for each CRL measurement.**

| CRL (mm) | PI | | S-wave(cm/s) | | a-wave(cm/s) | | TAMXV(cm/s) | |
|---|---|---|---|---|---|---|---|---|
| | P5 | P95 | P5 | P95 | P5 | P95 | P5 | P95 |
| 45 | 1.0007 | 1.3415 | 39.6783 | 65.5636 | 6.2287 | 16.6637 | 25.5151 | 46.1257 |
| 46 | 0.9960 | 1.3404 | 39.8012 | 65.7934 | 6.2531 | 16.8290 | 25.6328 | 46.3618 |
| 47 | 0.9915 | 1.3392 | 39.9343 | 66.0131 | 6.2796 | 16.9857 | 25.7579 | 46.5905 |
| 49 | 0.9830 | 1.3364 | 40.2310 | 66.4219 | 6.3394 | 17.2712 | 26.0302 | 47.0258 |
| 50 | 0.9790 | 1.3349 | 40.3946 | 66.6111 | 6.3727 | 17.3992 | 26.1774 | 47.2324 |
| 51 | 0.9751 | 1.3333 | 40.5684 | 66.7900 | 6.4085 | 17.5167 | 26.3320 | 47.4316 |
| 52 | 0.9715 | 1.3316 | 40.7524 | 66.9588 | 6.4467 | 17.6234 | 26.4939 | 47.6234 |
| 53 | 0.9680 | 1.3297 | 40.9466 | 67.1174 | 6.4874 | 17.7188 | 26.6633 | 47.8078 |
| 54 | 0.9647 | 1.3278 | 41.1509 | 67.2659 | 6.5308 | 17.8026 | 26.8400 | 47.9849 |
| 55 | 0.9617 | 1.3258 | 41.3654 | 67.4041 | 6.5768 | 17.8746 | 27.0241 | 48.1545 |
| 56 | 0.9588 | 1.3236 | 41.5901 | 67.5322 | 6.6256 | 17.9345 | 27.2157 | 48.3168 |
| 57 | 0.9562 | 1.3214 | 41.8250 | 67.6501 | 6.6773 | 17.9820 | 27.4145 | 48.4717 |
| 58 | 0.9538 | 1.3190 | 42.0701 | 67.7578 | 6.7320 | 18.0169 | 27.6208 | 48.6192 |
| 59 | 0.9516 | 1.3166 | 42.3253 | 67.8553 | 6.7898 | 18.0393 | 27.8345 | 48.7594 |
| 60 | 0.9496 | 1.3140 | 42.5907 | 67.9426 | 6.8508 | 18.0489 | 28.0555 | 48.8921 |
| 61 | 0.9478 | 1.3113 | 42.8663 | 68.0198 | 6.9152 | 18.0457 | 28.2839 | 49.0175 |
| 62 | 0.9463 | 1.3085 | 43.1521 | 68.0868 | 6.9830 | 18.0297 | 28.5197 | 49.1355 |
| 63 | 0.9451 | 1.3055 | 43.4480 | 68.1436 | 7.0545 | 18.0011 | 28.7629 | 49.2461 |
| 64 | 0.9440 | 1.3025 | 43.7542 | 68.1902 | 7.1298 | 17.9599 | 29.0134 | 49.3493 |
| 65 | 0.9432 | 1.2993 | 44.0705 | 68.2267 | 7.2090 | 17.9062 | 29.2714 | 49.4451 |
| 66 | 0.9427 | 1.2960 | 44.3970 | 68.2530 | 7.2924 | 17.8403 | 29.5367 | 49.5336 |
| 67 | 0.9424 | 1.2926 | 44.7337 | 68.2691 | 7.3803 | 17.7624 | 29.8094 | 49.6146 |
| 68 | 0.9423 | 1.2890 | 45.0805 | 68.2750 | 7.4727 | 17.6728 | 30.0895 | 49.6883 |
| 69 | 0.9425 | 1.2853 | 45.4376 | 68.2707 | 7.5699 | 17.5717 | 30.3770 | 49.7546 |
| 70 | 0.9430 | 1.2815 | 45.8048 | 68.2563 | 7.6722 | 17.4596 | 30.6718 | 49.8136 |
| 71 | 0.9436 | 1.2775 | 46.1822 | 68.2316 | 7.7799 | 17.3368 | 30.9741 | 49.8651 |
| 72 | 0.9446 | 1.2734 | 46.5697 | 68.1968 | 7.8933 | 17.2038 | 31.2837 | 49.9093 |
| 73 | 0.9457 | 1.2692 | 46.9675 | 68.1518 | 8.0127 | 17.0610 | 31.6007 | 49.9460 |
| 74 | 0.9471 | 1.2648 | 47.3754 | 68.0967 | 8.1385 | 16.9088 | 31.9251 | 49.9754 |
| 75 | 0.9488 | 1.2602 | 47.7935 | 68.0313 | 8.2710 | 16.7478 | 32.2569 | 49.9975 |
| 76 | 0.9506 | 1.2555 | 48.2218 | 67.9558 | 8.4106 | 16.5784 | 32.5960 | 50.0121 |
| 77 | 0.9527 | 1.2506 | 48.6603 | 67.8701 | 8.5580 | 16.4012 | 32.9425 | 50.0193 |
| 81 | 0.9633 | 1.2294 | 50.5160 | 67.4255 | 9.2346 | 15.6246 | 34.4025 | 49.9745 |
| 83 | 0.9699 | 1.2176 | 51.5049 | 67.1421 | 9.6344 | 15.2032 | 35.1768 | 49.9078 |
| 84 | 0.9734 | 1.2115 | 52.0146 | 66.9851 | 9.8525 | 14.9860 | 35.5750 | 49.8634 |

CHD using DV PI screening than qualitative DV assessment in NT thickened fetuses. Baran et al. [6] suggested that routine monitoring of the DV PI can provide valuable information on adverse pregnancy outcomes such as miscarriage, stillbirth, small for gestational age, low birth weight, fetal growth restriction, and severe congenital heart defects.

To further investigate the value of the DV PI in early pregnancy screening, it is necessary to establish a normal reference range. The DV PI P95 range in this study was from 1.3415 at CRL 45 mm to 1.2115 at CRL 84 mm, demonstrating a statistically significant inverse correlation with CRL. The DV S-wave, a-wave and TAMXV all increased with CRL, showing statistically significant correlations with CRL. This suggests a gradual increase in blood flow through the

DV with increasing gestational week, which may be due to trophoblast migration, disruption of the spiral arteries, decreased placental vascular resistance resulting in decreased cardiac afterload and increased cardiac compliance, decreased cardiac afterload resulting in increased blood flow velocity during atrial contraction, or a decreased pressure gradient between the umbilical vein and the right atrium causing the DV PI to decrease with gestational week [16–19].

Although the trend of DV blood flow velocity in this study was consistent with the results of previous studies, the DV PI reference range and trend of variation differed. Prefumo et al. [20] studied 198 singleton pregnancies and the mean DV PI ranged from 1.07 at 38 mm CRL to 1.00 at 88 mm CRL, showing a decreasing trend as CRL increased but the correlation with CRL values did not reach statistical significance. Teixeira et al. [21] observed a biphasic pattern of DV PI, with DV PI values increasing with CRL up to 63 mm CRL (equivalent to 12+6 weeks gestation) and decreasing thereafter. The authors suggested this may have been due to inadequate trophoblast migration, high cardiac afterload, or low blood flow velocity up to 12+6 weeks, resulting in high PI values in early pregnancy. In Brazilian and Thai populations [22, 23], no significant changes were observed in the DV PI values in relation to CRL values. The suggested reasoning for this was the immature establishment of the maternal-fetal circulation at this gestational age and the high resistance of the umbilical artery, which had an effect on fetal cardiac contractility as an afterload. These theories present some potential explanations for the changes in DV PI during pregnancy; however, further studies are required to validate and confirm these hypotheses. In addition, some studies are dated, and included different gestational weeks, CRL distributions, races, and genetic and environmental backgrounds, which may have affected the outcomes and resulted in different DV PI reference ranges and trends.

In recent years, advanced ultrasound equipment and continuous improvements in ultrasound technology have made DV Doppler flow measurements more standardised and effective at avoiding interference from peripheral vessels, and automatic waveform tracing techniques have increased the objectivity of the values. Furthermore, Sabria et al. [24] concluded that the DVPI values of 14,444 singleton fetuses measured between11+0 to 13+6 weeks exhibited changes over the course of four years. These changes were attributed to the updates in ultrasound equipment and the development of new technologies during this period.

Our sample selection was more rigorous than in previous studies and all cases were followed up during pregnancy and the perinatal period. Those with fetal structural abnormalities, chromosomal abnormalities, induction of abortion, loss to follow-up, or poor pregnancy outcome were excluded at follow-up, as were those with maternal pregnancy complications such as hypertension and diabetes mellitus. All examinations were performed by a single FMF-certified sonographer with more than 10 years' experience in prenatal ultrasound diagnosis, which reduces interobserver error and increases the accuracy and reliability of our results.

In conclusion, the DV PI value is an important screening tool in early pregnancy. Therefore, the reference range established in this study will serve as a valuable tool for detecting fetal chromosomal abnormalities and congenital heart defects, as well as predicting other adverse pregnancy outcomes. The limitation of this study is the small number of cases, and studies involving a larger sample size are required to establish a high-quality normal reference range for DV PI. Further studies on the clinical value of normal and abnormal DV PI values in relation to adverse pregnancy outcomes will be conducted on this basis.

## Supporting information

**S1 Fig. Fetal DV PI 5th, 50th and 95th percentile measurements in 224 fetuses.**
(TIF)

**S2 Fig. Fetal DV S-wave 5th, 50th and 95th percentile measurements in 224 fetuses.**
(TIF)

**S3 Fig. Fetal DV a-wave 5th, 50th and 95th percentile measurements in 224 fetuses.**
(TIF)

**S4 Fig. Fetal DV TAMXV 5th, 50th, and 95th percentile measurements in 224 fetuses.**
(TIF)

**S1 File. Raw data of 224 fetuses.**
(XLSX)

## Author Contributions

**Conceptualization:** Shizhen Li, Xiangyi Dong.

**Formal analysis:** Shizhen Li, Qi Li, Xiangyi Dong.

**Investigation:** Shizhen Li, Linlin Zhu, Xiangyi Dong.

**Methodology:** Shizhen Li, Haifang Wu, Linlin Zhu, Qi Li, Xiangyi Dong.

**Project administration:** Shizhen Li, Xiangyi Dong.

**Resources:** Shizhen Li, Haifang Wu, Linlin Zhu, Qi Li, Xiangyi Dong.

**Supervision:** Shizhen Li, Xiangyi Dong.

**Writing – original draft:** Shizhen Li, Haifang Wu, Xiangyi Dong.

**Writing – review & editing:** Shizhen Li, Haifang Wu, Linlin Zhu, Qi Li, Xiangyi Dong.

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
