## [Decision Letter · Decision Letter 0]

22 Aug 2024

PONE-D-24-27078Reference values of normal fetal ductus venosus Doppler flow measurements at 11–14 weeks of gestationPLOS ONE

Dear Dr. Dong,

Thank you for submitting your manuscript to PLOS ONE. After careful consideration, we feel that it has merit but does not fully meet PLOS ONE’s publication criteria as it currently stands. Therefore, we invite you to submit a revised version of the manuscript that addresses the points raised during the review process. Please submit your revised manuscript by Oct 06 2024 11:59PM. If you will need more time than this to complete your revisions, please reply to this message or contact the journal office at plosone@plos.org. Please include the following items when submitting your revised manuscript:A rebuttal letter that responds to each point raised by the academic editor and reviewer(s). You should upload this letter as a separate file labeled 'Response to Reviewers'.A marked-up copy of your manuscript that highlights changes made to the original version. You should upload this as a separate file labeled 'Revised Manuscript with Track Changes'.An unmarked version of your revised paper without tracked changes. You should upload this as a separate file labeled 'Manuscript'.If applicable, we recommend that you deposit your laboratory protocols in protocols.io to enhance the reproducibility of your results. Protocols.io assigns your protocol its own identifier (DOI) so that it can be cited independently in the future. For instructions see: https://journals.plos.org/plosone/s/submission-guidelines#loc-laboratory-protocols. Additionally, PLOS ONE offers an option for publishing peer-reviewed Lab Protocol articles, which describe protocols hosted on protocols.io. Read more information on sharing protocols at https://plos.org/protocols?utm_medium=editorial-email&utm_source=authorletters&utm_campaign=protocols.

We look forward to receiving your revised manuscript.

Kind regards,

Zahra Hoodbhoy

Academic Editor

PLOS ONE

Journal Requirements:

Reviewers' comments:

Reviewer's Responses to Questions

**Comments to the Author**

1. Is the manuscript technically sound, and do the data support the conclusions?

Reviewer #1: Yes

Reviewer #2: Yes

2. Has the statistical analysis been performed appropriately and rigorously? 

Reviewer #1: Yes

Reviewer #2: Yes

3. Have the authors made all data underlying the findings in their manuscript fully available?

Reviewer #1: Yes

Reviewer #2: Yes

4. Is the manuscript presented in an intelligible fashion and written in standard English?

Reviewer #1: Yes

Reviewer #2: Yes

5. Review Comments to the Author

Reviewer #1: The introduction is well-structured and covers the necessary background information effectively.

The timeframe and setting are clearly stated. However, it would be beneficial to specify the hospital's location for context. Additionally, consider mentioning the number of participants (if available) to give readers a better sense of the study's scale.Doppler Ultrasonography (Lines 76-94):This is a crucial part of the methodology. It might be helpful to specify whether the angle of <30° is based on specific guidelines or studies. Additionally, the process of repeating measurements and taking the mean value is good practice, but it could be elaborated on to explain why three repetitions were chosen.

Lines 113-120 (Study Population):This is a clear and concise summary of the final study population. The inclusion of demographic data (mean maternal age, BMI, etc.) is useful. However, it might be helpful to specify what "median gestation and delivery numbers" refer to—whether it is the number of pregnancies and deliveries or another metric.

Lines 122-124 (DV PI Results):The results are presented clearly, and the statistical significance is noted. However, the sentence is somewhat dense. It may be helpful to split it: "For the 224 normal fetuses, the DV PI P5 and P95 ranged from 1.0007 and 1.3415 for a CRL of 45 mm to 0.9734 and 1.2115 for a CRL of 84 mm. This correlation with CRL values was statistically significant (Pearson's r = -0.163; P = 0.015).

in the end put the section of limitations or weakness and write a following

Weaknesses:

Sample Size: The study might benefit from a larger sample size to enhance the statistical power and generalizability of the findings.

Single-Center Study: The research appears to be conducted at a single center. Multi-center studies could provide more diverse data and validate the results across different populations.

Limited Gestational Range: The study focuses on the 11–14 weeks of gestation period. Expanding the gestational range could provide a more comprehensive understanding of fetal ductus venosus flow dynamics.

Reviewer #2: well written provide quantitative assessment in addition to qualitative. Though qualitative assessment of Ductus Venosus flow on its own is a pretty good marker to add up to the existing aneuploidy screening tools.

My Question: How much weightage would you give to addition of quantification of ductus in addition to the overall screening methods?

6. PLOS authors have the option to publish the peer review history of their article (what does this mean?). If published, this will include your full peer review and any attached files.

Reviewer #1: **Yes: **Shazia Mohsin

Reviewer #2: **Yes: **Zaheena Shamsul Islam

---

## [Author Response · Author response to Decision Letter 0]

30 Sep 2024

We have carefully revised the manuscript based on the valuable suggestions from the editor and reviewers. All specific issues have been addressed in our reply letter to the reviewers. We sincerely appreciate the time and effort that the editor and reviewers have dedicated to reviewing our manuscript.

Reviewer #1: 

The timeframe and setting are clearly stated. However, it would be beneficial to specify the hospital's location for context. Additionally, consider mentioning the number of participants (if available) to give readers a better sense of the study's scale. 

Thank you for your meticulous review and valuable suggestions. We have added the hospital's address into the article. Our study encompassed a total of 343 fetuses; however, due to incomplete follow-up data, some were not included in the article.

Doppler Ultrasonography (Lines 76-94): This is a crucial part of the methodology. It might be helpful to specify whether the angle of <30° is based on specific guidelines or studies.

The Certification training section on Ductus venosus flow on the British Fetal Medicine Foundation (FMF) training website states that "The insonation angle should be less than 30 degrees." （https://fetalmedicine.org/fmf-certification-2/ductus-venosus-flow）We have taken this criterion into consideration.

Additionally, the process of repeating measurements and taking the mean value is good practice, but it could be elaborated on to explain why three repetitions were chosen.

We consulted existing literature and calculated the average of three measurements to enhance accuracy. Considering that conducting measurements only once or twice may lead to greater errors, while excessive measurements could extend examination duration, we opted for averaging three measurements in accordance with the ALARA (As Low As Reasonably Achievable) principle for fetal ultrasound examinations. This approach aims to achieve optimal measurement precision within a minimized examination timeframe

Lines 113-120 (Study Population): This is a clear and concise summary of the final study population. The inclusion of demographic data (mean maternal age, BMI, etc.) is useful. However, it might be helpful to specify what "median gestation and delivery numbers" refer to—whether it is the number of pregnancies and deliveries or another metric.

Yes, this refers to the median number of pregnancies and deliveries. Thank you for your correction. The expression has been modified in the text.

Lines 122-124 (DV PI Results): The results are presented clearly, and the statistical significance is noted. However, the sentence is somewhat dense. It may be helpful to split it: "For the 224 normal fetuses, the DV PI P5 and P95 ranged from 1.0007 and 1.3415 for a CRL of 45 mm to 0.9734 and 1.2115 for a CRL of 84 mm. This correlation with CRL values was statistically significant (Pearson's r = -0.163; P = 0.015).

Thank you for your modification suggestions. We have implemented the changes as per your advice in the main text to render the sentence more succinct and lucid.

Weaknesses:

Sample Size: The study might benefit from a larger sample size to enhance the statistical power and generalizability of the findings.

Single-Center Study: The research appears to be conducted at a single center. Multi-center studies could provide more diverse data and validate the results across different populations.

The single-center design and the relatively small sample size are indeed the deficiencies of this study. We are highly grateful to the reviewer for highlighting this issue. Please grant us an opportunity to elaborate on these limitations. The single-center study was conducted as we aimed for better quality control, to minimize measurement errors introduced by different measurement techniques among different hospitals and personnel, and to ensure accurate follow-up results. Thus, we initiated our research with a single-center study as the first step. Our sample size was relatively limited; however, during the research process, we observed some abnormal cases and discovered that an increase in DVPI might be associated with adverse pregnancy outcomes. We aspire to publish an article on the normal reference range of DVPI to assist more physicians in providing clinical consultation and guidance for pregnant women. Due to the relatively small number of abnormal cases discovered and the fact that we are still in the process of accumulating cases, abnormal cases are not discussed in this paper. We will undertake multi-center studies with a larger sample size in the subsequent process.

Limited Gestational Range: The study focuses on the 11–14 weeks of gestation period. Expanding the gestational range could provide a more comprehensive understanding of fetal ductus venosus flow dynamics.

Further research on DVPI across various gestational weeks is certainly important. However, our study specifically concentrates on the early pregnancy screening period. The ISUOG guidelines recommend that qualitative assessment of DV be routinely performed during this stage, as it plays a vital role in aneuploidy screening. Additionally, the latest ISUOG guidelines suggest that elevated DVPI may correlate with increased detection rates of aneuploidy. Consequently, we examined the normal values of DVPI at 11-14 weeks. Abnormalities in DVPI during mid and late pregnancy could have unique clinical implications, such as fetal growth restriction, which also deserves attention; however, this topic falls outside the current paper's scope. We sincerely appreciate the reviewers for providing us with valuable suggestions for future research directions. Multi-center studies, larger sample sizes, and investigations into normal values of DVPI during mid and late pregnancy are potential avenues we look forward to exploring.

Reviewer #2: well written provide quantitative assessment in addition to qualitative. Though qualitative assessment of Ductus Venosus flow on its own is a pretty good marker to add up to the existing aneuploidy screening tools.

My Question: How much weightage would you give to addition of quantification of ductus in addition to the overall screening methods?

Thank you for raising such an insightful question. We took the time to review the fetuses that were excluded from our study due to structural malformations or chromosomal abnormalities. Among these, we identified five cases with normal NT and a positive DV a wave, yet they exhibited abnormally elevated PI values. Specifically, one case was confirmed as trisomy 21 through amniocentesis, another presented with a diaphragmatic hernia along with increased echogenicity of both kidneys, one had an atrioventricular septal defect also confirmed as trisomy 21 by amniocentesis, one showed a single umbilical artery, and another unfortunately experienced intrauterine demise at 16 weeks of gestation. Therefore, we believe that the increase in DVPI during early pregnancy is not only linked to an elevated risk of aneuploidy but may also be associated with adverse pregnancy outcomes like structural anomalies. However, given the limited sample size, we did not delve into discussing the cases of abnormally high DVPI and their implications within this article; instead, we focused on presenting the normal values of DVPI for reference by other researchers. We are actively continuing to accumulate more cases and look forward to uncovering additional insights regarding quantitative assessment of DV in future studies.

---

## [Decision Letter · Decision Letter 1]

15 Oct 2024

Reference values of normal fetal ductus venosus Doppler flow measurements at 11–14 weeks of gestation

PONE-D-24-27078R1

Dear Dr. Dong,

We’re pleased to inform you that your manuscript has been judged scientifically suitable for publication and will be formally accepted for publication once it meets all outstanding technical requirements.

Kind regards,

Zahra Hoodbhoy

Academic Editor

PLOS ONE

---

## [Editor Report · Acceptance letter]

17 Oct 2024

PONE-D-24-27078R1 

PLOS ONE

Dear Dr. Dong, 

I'm pleased to inform you that your manuscript has been deemed suitable for publication in PLOS ONE. Congratulations! Your manuscript is now being handed over to our production team.

Kind regards, 

on behalf of

Dr. Zahra Hoodbhoy 

Academic Editor

PLOS ONE